# Nutritional Assessment in Gastrointestinal Tumors: News from the 2020 ASCO and ESMO World GI Meetings

**Federica Mascaretti [1],\* and Jessica Evangelista [2]**

[1] Unità di Gastroenterologia ed Endoscopia, Fondazione IRCCS Ca' Granda Ospedale Maggiore Policlinico, 20019 Milano, Italy

[2] Unità Operativa complessa di Chirurgia Toracica, Dipartimento di Scienze Mediche e Chirurgiche, Fondazione Policlinico Universitario Agostino Gemelli IRCCS, 00100 Rome, Italy; evangelistajessica664@gmail.com

\* Correspondence: mascaretti.federica@nutritiongoals.eu; Tel.: +39-3338-512948

**Abstract:** Nutritional risk screening and assessment of general nutritional status are of primary importance in the management of gastrointestinal cancers (GIC). Indeed, a major problem in these patients is the involuntary weight loss leading eventually to cachexia. With our review, we aimed at collecting the most recent advances in nutritional assessment of patients with GIC. All the abstracts presented both at the 2020 ASCO and ESMO World GI meetings were considered and a total number of 12 abstracts were selected, reporting colorectal, gastric, esophageal, and pancreatic cancer (PC) series. In some of the analyzed series, pathological conditions such as cachexia and sarcopenia had prognostic significance on clinical outcomes. One abstract reported the results of a phase I trial with the use of a novel interleukin-1-alpha antagonist, bermekimab. Its association with standard chemotherapy in advanced PC brought an improved patients' performance during treatment. Insufficient attention is paid to the nutritional status of patients with GIC both at screening and during specific cancer treatment. The use of antropometric measurements, together with nutritional assessment tools, may facilitate the clinical evaluation of these patients. Large randomized trials are warranted in order to clarify the real impact of nutritional interventions on clinical outcomes.

**Keywords:** nutrition; gastrointestinal cancers; cachexia; BMI; malnutrition; gastric cancer; colorectal cancer; esophageal cancer; pancreatic cancer

## 1. Introduction

Gastrointestinal cancers (GIC) include a wide number of primary tumors: esophagus, stomach, liver, gallbladder, pancreas, small intestine, colon and rectum, and anus [1]. GIC account for 26% of new diagnoses of cancer and 35% of all cancer-related deaths. In 2018, there were 4.8 million estimated new cases of GIC and 3.4 million related deaths worldwide [2]. The majority of cases and deaths occurred in Asia (63 and 65%, respectively), followed by Europe and North America, accounting together for 26% of total cases and 23% of deaths [2].

More than a half of all GIC are associated to modifiable risk factors such as smoking, alcohol consumption, infections, central obesity, and exposure to radiations [1,2]. Diet directly affects GIC risk, as well. Indeed, it has been shown that the excessive consumption of salt-preserved food and processed meat is associated with an increase in GIC incidence. On the other hand, fiber, fruit and vegetable consumption has a protective role against GIC onset [3].

Nutritional risk screening and assessment of general nutritional status are of primary importance in the management of GIC. Indeed, a major problem in patients with GIC is the involuntary weight loss

leading eventually to cachexia development [1]. The prevalence of cachexia is very high in patients with pancreatic and gastric cancer, being as high as 87%, while it can reach a 61% rate in colorectal cancer. Cancer cachexia is a complex syndrome characterized by autophagy with a progressive decline of skeletal muscle, with or without fat loss. It leads to worsening functional impairment, reduced efficacy of chemotherapy (CT), increased toxicities, and a death rate up to 20% of all cancer-related deaths [4].

Considering the influence of nutritional status on better clinical outcomes in GIC, early nutritional screening is suggested in order to identify risk of malnutrition. Some screening tools have been validated and are reported in Table 1.

**Table 1.** Screening instruments in nutrition.

| Screening Instruments | Main Measures |
|---|---|
| Malnutrition Screening Tool | Unintentional weight loss, appetite |
| Nutritional Risk Screening 2002 | Weight loss, BMI, food intake |
| Malnutrition Universal Screening Tool | BMI, weight changes |

All of them include either the evaluation of weight loss or the calculation of body mass index (BMI). In addition, some tools for nutritional assessment are available and can be used during the clinical history of cancer patients (Table 2) [1].

**Table 2.** Nutritional assessment tools.

| Screening Instruments | Main Measures |
|---|---|
| Subjective Global Assessment (SGA) | Weight loss, food intake, GI symptoms, functional capacity |
| Patient-Generated SGA (PG-SGA) | Weight loss, food intake, GI symptoms, functional capacity |
| Revised Mini Nutritional Assessment Short Form | BMI or calf circumference, weight loss, appetite |
| Prognostic Nutritional Index (PNI) | Serum albumin, blood lymphocyte count |

As the Subjective Global Assessment (SGA) and the Patient-Generated Subjective Global Assessment (PG-SGA) combine medical history (weight change, dietary intake change, gastrointestinal symptoms and functional capacity changes) together with physical changes (subcutaneous fat loss, muscle wasting, ascites), they are considered to be the most useful. They both combine data from medical history (with physical changes). However, while the first tool is completed by health professionals only, the latter can be completed by patients and adheres more to nutritional changes happening over a short period of time [5]. Another important tool is the prognostic nutritional index (PNI). The modified PNI suggested by Onodera et al. includes two factors: serum albumin and blood lymphocyte count [6]. It is a simple tool used for assessment of preoperative conditions, prediction of short-term postoperative complications, and evaluation of long-term outcomes in many cancer types including GC [7]. Low values of PNI have been associated with poor overall survival (OS) in gastric cancer [8].

This review aims at collecting the most recent advances in nutritional assessment of patients with GIC. In particular, the latest abstracts presented at the 2020 American Society of Clinical Oncology (ASCO) and European Society of Medical Oncology World Congress on Gastrointestinal Cancer (ESMO World GI) meetings are included and critically discussed.

## 2. Results

A total number of 12 abstracts were selected. Five abstracts were selected from the 2020 ESMO World GI meeting while seven were selected from the 2020 ASCO Meeting. Four abstracts reported colorectal cancer (CRC) series; two abstracts reported gastric cancer (GC) series; in one, both gastric and CRC patients' outcomes were reported, two abstracts concerned esophageal cancer (EC), while three abstracts treated pancreatic cancer (PC) patients' outcomes (Table 3).

**Table 3.** Summary of the selected abstracts.

| Article Reference | Geographical Origin | Index | Results |
|---|---|---|---|
| **Colorectal Cancer** | | | |
| Costa et al., 2020 [9] | Portugal | PNI | Low PNI: worst results in stage III CRC patients submitted to curative surgery and adjuvant chemotherapy (CT). |
| Treasure et al., 2020 [10] | USA | GL | CRC patients with a low glycemic index diet:<br>- Decreased BMI and waist circumference<br>- Decreased total cholesterol, VLDL and triglycerides<br>- Weight loss. |
| da Silva Dias et al., 2020 [11] | Portugal | BMI | Low BMI at diagnosis:<br>- Worse prognosis<br>- Increased toxicities. |
| Gallois et al., 2020 [12] | France | SMI | Decrease in SMI > 14% during first 2 months of treatment was associated with decreased OS. |
| Goncalves et al., 2020 [13] | Brazil | SIBO and lactose intolerance | Diarrhea caused by CT treatment for CRC or gastric cancer was independent of the presence of SIBO or lactose intolerance. |
| **Gastric Cancer** | | | |
| Aizawa et al., 2020 [14] | Japan | PNI | Low PNI related to recurrence after GC resection in the locally advanced disease. |
| Cipriano et al., 2020 [15] | Portugal | NLR | - NLR was associated with poor outcomes.<br>- No differences in OS taking into account PNI and BMI. |
| **Esophageal Cancer** | | | |
| Squires et al., 2020 [16] | USA | Weight loss | - Median 6-month post-esophagectomy weight loss was 10%.<br>- Greater preoperative BMI associated with increased weight loss. |
| Soriano et al., 2020 [17] | Canada | SMI, VAT, SAT | - Sarcopenia and low VAT resulted in worse OS.<br>- Sarcopenia increased the risk of death by 50%. |
| **Pancreatic Cancer** | | | |
| Trestini et al., 2020 [18] | Italy | BC | - Sarcopenia was related to higher complications rate after surgery (e.g., cardiac complications).<br>- Patients with postoperative complications had higher preoperative total adipose tissue.<br>- Sarcopenic obesity at baseline was found to be a significant independent predictor for OS. |
| Appleyard et al., 2020 [19] | United Kingdom | PNI | PNI associated with OS. |
| Atkins et al.; 2020 [20] | USA | BC | Bermekimab, nano-liposomal irinotecan and 5-fluorouracil in refractory pancreatic cancer patients with cachexia:<br>- Fat-free mass and fat mass change (−1.6 kg and −1.4 kg).<br>- Improved well-being. |

*2.1. Colorectal Cancer*

In a retrospective series of 235 patients with stage III CRC subjected to curative surgery and adjuvant CT, the PNI score was evaluated and correlated with chronic inflammatory parameters such as

total lymphocyte count and lymphocyte-to-monocyte ratio (LMR). PNI was calculated according to the modified Onodera's formula: serum albumin level (g/L) + 5x total lymphocyte count (/L). The cut-off value of PNI was 51.70, with the majority of patients (51.9%) having low PNI. OS and disease-free survival (DFS) were significantly lower in the low PNI group than in the PNI > 51.70 cohort (90.9 vs. 95.4%, $p = 0.014$ for OS, 72.7 vs. 87.3%, $p = 0.007$). In the multivariate analysis, PNI ≤ 51.70 maintained a negative impact only on DFS (HR 1.457; CI 95% 1.215–1.749; $p < 0.001$). PNI was positively correlated with total lymphocyte count ($r = 0.707$, $p < 0.001$) and with LMR ($r = 0.480$, $p < 0.001$), while it was negatively associated with neutrophil-to-lymphocyte ratio [NLR ($r = -0.394$, $p < 0.001$)] and with platelet-to-lymphocyte ratio [PLR ($r = -0.410$, $p < 0.001$)] [9]. Therefore, low PNI was found to be associated with poorer survival outcomes in patients treated with surgery and adjuvant CT for localized CRC. Moreover, low PNI values were positively associated with systemic inflammatory response parameters. In the same setting of patients with resectable stage I-III CRC, a cohort of 18 patients with an average glycemic load (GL) > 150 participated in a dietary intervention with a target GL ≤ 102. A food acceptability questionnaire was administered monthly. Moreover, patients were tested for laboratory parameters such as total cholesterol, VLDL cholesterol and triglycerides. Most of the patients (67%) resulted in being compliant with a low GL ≥ 75% of the time and had a reduction both in BMI and waist circumference. Moreover, 28% experienced a weight loss ≥ 5%. Significant decreases were seen in total cholesterol (7.2% decrease; $t = -2.33$, $p = 0.03$), VLDL (26.8% decrease; $t = -2.33$, $p = 0.03$) and triglycerides (26.6% decrease; $t = -2.29$; $p = 0.04$). All participants were satisfied with the diet with 43% being extremely satisfied. This study was the first step in designing a larger scale trial in order to possibly evaluate the impact of low GL diet on cancer outcomes [10].

In patients with advanced CRC, a retrospective analysis of 178 patients treated with first line CT plus biological agent (either anti-VEGF or anti-EGFR) was performed in order to evaluate the impact of nutritional intervention on BMI, treatment toxicities and survival outcomes. The mean BMI was 24.8 Kg/m$^2$. A negative mean BMI difference (BMIdif) was observed after progression with both therapies. There was no statistically significant association between BMIdif and OS, but there was a trend towards BMIdif and severe cutaneous toxicity ($p = 0.033$; OR 1.99; IC 95% [1.05–1.49]), and anti-EGFR discontinuation ($p = 0,044$; OR 2.33; CI 95% [1.02–5.31]). A rate of 17,6% of patients was referred to nutritional intervention, with a positive association with BMIdif after gaining weight [11]. In the same disease setting of advanced and non-pretreated CRC, a cohort of 149 patients was analyzed prospectively in order to evaluate the association between baseline sarcopenia and the variation of the Skeletal Muscle Index (SMI) under treatment with survival and treatment-related toxicities. SMI was measured with computed tomography scan at day 0 and day 60. Sarcopenia at baseline was not significantly associated with survival outcomes or CT toxicities. A decrease in SMI > 14% between day 0 and 60 was significantly associated with shorter progression-free survival (PFS) (6 vs. 9 months; HR 1.8, 95%CI 1.1–3.1, $p = 0.02$) and OS (8.5 vs. 26 months; HR 2.4, 95%CI 1.3–4.4, $p = 0.004$) in multivariate analysis. A significant percentage of 40% of patients with a SMI decrease > 14%, and 22% of patients with a SMI increase or stable or decrease < 14% developed grade ≥ 2 clinical toxicities (OR 3.0, 95%CI 1.2–7.7, $p = 0.02$). The difference, however, was not statistically significant in multivariate analysis (OR 2.3. 95%CI 0.8–6.7, $p = 0.1$). Thus, baseline sarcopenia was not associated with poor survival outcomes, but the decrease in SMI > 14% during the first 2 months of treatment was associated with decreased survival, independently of other prognostic and nutritional factors such as hypoalbuminemia and malnutrition measured with PG-SGA [12].

A different abstract evaluated the small intestine bacterial overgrowth (SIBO) and lactose intolerance incidence in 33 patients with CRC ($n = 29$) and GC ($n = 3$). Most of the patients underwent CT with a fluoropyrimidine (5FU or capecitabine) and oxaliplatin (54.5%). SIBO and lactose intolerance were then correlated with nutritional status and presence of diarrhea. To detect SIBO and lactose intolerance, the expired hydrogen test was used. The number and aspects of the evacuations and toxicity degree were collected. For the nutritional assessment, both the BMI and PG-SGA scores were calculated. Diarrhea was present in 57.5% cases, in 13 cases (39.4%) it was of grade II/III. According

to PG-SGA, 84.9% had moderate or severe nutritional risk grade. A percentage of 45% had lactose intolerance and 9% SIBO. Despite the low number of patients considered, no relationship was found between diarrhea, SIBO and lactose intolerance. Therefore, diagnosis of other etiologies of diarrhea may contribute to better tolerance of treatment and improved quality of life [13].

## 2.2. Gastric Cancer

A series of 567 patients undegoing radical resection for GC were evaluated in order to identify postoperative changes of PNI. The PNI was calculated as $10 \times$ serum albumin concentration (g/dL) + $0.005 \times$ peripheral total lymphocyte count (/μL). The postoperative laboratory data and PNI assessment were evaluated at postoperative day (POD) 1, 3, 7, and first visit after discharge (AD). The predictive value of low PNI-AD on the relapse free survival (RFS) after surgery was estimated to be < 40. The median value of PNI at base line, 1, 3, 7 POD, and AD was 51.0, 35.9, 34.0, 38.3 and 48.2, respectively. The PNI significantly decreased at 1, 3 and 7 POD. Though the PNI at AD had an increase and proclivity to recover, it was still significantly lower than that at baseline. At the multivariate analysis, PNI-AD < 40 (HR: 1.833, 95%CI; 1.002–3.352, $p = 0.04$) was found to be an independent predictor of recurrence, together with female gender, pT3-4, pN1-3, and vessel invasion. In conclusion, low PNI-AD was significantly related to recurrence after GC resection in the locally advanced disease (pT3-4, pN1-3, stage III) but not in stage IB and II [14].

In the metastatic disease, 55 patients receiving CT for advanced GC were considered in order to evaluate prognostic factors. NLR was ≥ 5 in 34.5% ($n = 19$) and PNI ≥ 50 in 21.8% ($n = 12$). BMI was <5, $p = 0.009$. At the multivariate analysis, NLR ≥ 5 was associated with a poor OS (HR 2.28; 95%CI 1.15–4.52; $p = 0.019$). In addition, metachronous metastatic disease and previous perioperative/adjuvant CT were associated with poor outcomes. On the contrary, there were no statistically significant differences in OS taking into account sex, age, ECOG PS, tumor location, histology, peritoneal carcinomatosis, number of sites of metastatic disease, Ca19-9, CEA, albumin, PNI, and BMI [15].

## 2.3. Esophageal Cancer

A group of 176 patients undergoing minimally-invasive esophagectomy for EC were evaluated in order to understand the risk factors associated with post-esophagectomy weight loss. The median 3-month postoperative weight loss was 7.9% [interquartile range (IQR) 1.5–12.3%], while the median 6-month postoperative weight loss was 10% [IQR 5.3–15.0%]. On multivariable analysis, greater preoperative BMI ($p = 0.045$ at 3 months, $p = 0.007$ at 6 months post-surgery) and anastomotic leak ($p = 0.003$ at 3 months, $p = 0.020$ at 6 months) were associated with increased weight loss [16].

In the metastatic disease, a cohort of 200 patients was considered to determine Skeletal Muscle Index (SMI), Visceral Adiposity Tissue (VAT), and Subcutaneous Adiposity Tissue (SAT) with baseline computed tomography scan. We found that 104 (52%) were sarcopenic at baseline, 66 (33%) had high VAT, and 67 (34%) had high SAT. Multivariate model showed that sarcopenia and VAT were independent prognostic variables for 3-year OS. Sarcopenia increased the risk of death by 50% (HR:1.50, $p = 0.02$), whereas every 100-cm$^2$ increase in VAT improved OS by 24% (HR:0.76, $p = 0.03$). Moreover, at evaluation of FACT-E questionnaires, sarcopenic patients had significantly worse physical well-being ($p = 0.01$) after adjusting for sex and age [17].

## 2.4. Pancreatic Cancer

One of the selected abstracts evaluated the impact of sarcopenia/sarcopenic obesity and changes in body composition (BC) following neoadjuvant CT in patients with resectable PC. The final cohort consisted of 108 patients, with 91 patients (89.8%) who received neoadjuvant CT with FOLFIRINOX. Sarcopenia and sarcopenic obesity were found at diagnosis in 41.7% and 16.7% of patints, respectively. After FOLFIRINOX, the prevalence of sarcopenia and sarcopenic obesity decreased to 30.5% and 10.2%, respectively. Sarcopenia and sarcopenic obesity were both related to higher overall complications rate after surgery. In particular, cardiac complications were associated with sarcopenia (9% vs. 0%,

$p = 0.028$) and patients with postoperative complications had higher preoperative total adipose tissue (300 vs. 229 cm$^2$, $p > 0.05$). At multivariate analysis, sarcopenic obesity at baseline (HR 3.45, $p = 0.023$) was found to be a significant independent predictor for OS [18].

In 138 patients with locally advanced inoperable or metastatic PC that received FOLFIRINOX as first-line treatment, different clinical biomarkers were evaluated for their potential prognostic significance. NLR, monocyte-to-lymphocyte ratio (MLR), PLR, PNI, and systemic inflammation response index (SIRI), calculated as: (neutrophils * monocytes)/lymphocytes. NLR (HR 1.08, 95%CI 1.04–1.11, $p < 0.001$), MLR (HR 7.57, 95%CI 3.05–18.83, $p < 0.001$), PLR (HR 1.004, 95%CI 1.002–1.006, $p < 0.001$), SIRI (HR 1.12, 95%CI 1.07–1.17, $p < 0.001$), and PNI (HR 0.97, 95%CI 0.94–0.99, $p\frac{1}{4}0.011$) were all associated with OS. However, while NLR, MLR, PLR, and SIRI were associated with poor OS independent of age, sex, PS ECOG, Charlson Comorbidity Index (CCI) and stage (metastatic versus locally advanced), PNI did not demonstrate independent prognostic significance [19]. In another series, patients with advanced PC suffering from cachexia were included in a phase I study of second-line treatment (after progression to gemcitabine-based treatment). In this trial, standard CT with nanoliposomal irinotecan and 5-fluorouracil/folinic acid was combined with interleukin-1-alpha antagonist bermekimab. In total, 21 patients were enrolled and 18 were evaluable. Bermekimab in combination with nanoliposomal irinotecan (70 mg/m$^2$) and 5-fluorouracil (2400 mg/m2) was well tolerated at the highest dose level (12 mg/kg). Fat-free mass and fat mass change was −1.6 kg (± 2.0; $p = 0.003$) and −1.4 kg (± 1.7; $p = 0.004$), respectively. Reactive-C protein was 20.4 (± 35.6) at cycle 1 (C1) and decreased significantly ($p = 0.005$). Serum vascular endothelial growth factor decreased from C1 to cycle 3 ($p = 0.007$). Average daily step counts increased by 589 steps/day ($p = 0.29$) and resting heart rate decreased by 2.5 beats per minute ($p = 0.005$). Moreover, quality-of-life questionnaires reported an improvement for functional well-being. As a whole, the addition of bermekimab to standard CT was well-tolerated and improved the patients' performance during treatment [20].

## 3. Discussion

Even though the importance of nutritional status in cancer care is stressed with position papers and guidelines, its appropriate consideration is often lacking. Certainly, randomized controlled trials are scarce and, historically, advances in cancer care have been focused on the introduction of new substances [21]. Recently, in an Italian series including almost 2000 cancer patients, the reported prevalence of nutritional impairment was 51%, with 9% of patients clearly malnourished [22]. Moreover, a French study including 1903 cancer patients reported a 39% rate of malnutrition prevalence. In more than 40% of patients identified as malnourished, there was no nutritional support provided [23]. Certainly, among oncologists, an important under-recognition of cancer-related nutritional impairment has been reported, as well. In our review, we analyzed series concerning cancer patients with cachexia. Cachexia is a multifactorial syndrome characterized by involuntary weight loss, ongoing loss of skeletal muscle mass with or without loss of fat mass [21]. In the discussed phase I studies including PC cancer that progressed to a first-line CT, cachexia was treated with a specific drug, acting on the mechanism of systemic inflammation underlying this condition [20]. Differently, sarcopenia is described as a progressive and generalized loss of skeletal muscle mass and function potentially associated with falls, fractures, physical disability, and mortality. Malnutrition may be a cause of secondary sarcopenia in cancer [21]. Standards for the measurement of sarcopenia are based on computed tomography of muscle mass. Conventionally, cutoffs for skeletal muscle measurement at the level of the third lumbar vertebra (L3) are used (SMI). However, L3 evaluation is not always included in computed tomography scans. For these cases, alternative levels may be L2-L4-L5-L1 or thoracic vertebrae T12-T11-T10 [24]. Sarcopenia has been described as a negative prognostic factor in patients with PC undergoing endoscopic ultrasound-guided celiac plexus neurolysis [25]. Moreover, a decrease in SMI within 2 months after the initiation of treatment with FOLFIRINOX for PC was associated with a significantly shorter OS [26]. Sarcopenia was found to be associated with poorer OS in CRC, EC and PC. In particular, a significant reduction in the SMI during CT was associated with poorer survival in

advanced CRC [12], while basal sarcopenia increased the risk of death of 50% in metastatic EC [17]. In resectable PC, sarcopenic obesity at baseline was an independent negative predictor of OS [18].

Guidelines suggest regular screening at cancer diagnosis, in order to detect malnutritional disturbances. Weight changes and BMI evaluation are of fundamental importance in this process [11,16,27]. In the case of abnormal screening, evaluation of the nutritional intake, nutrition impact symptoms, muscle mass, physical performance, and the degree of systemic inflammation is strongly suggested. Available tools are various with no widespread scientific consensus on them. In particular, specific grading of the reported deficits is missing [27]. In the analyzed abstracts, both PNI [9,14] and PG-SGA found application [12,13].

In conclusion, a deeper attention should be paid to the nutritional status of patients with GIC both at screening and during specific cancer treatment. Clinical evaluation of anthropometric measurements, together with the use nutritional assessment tools, may facilitate the nutritional evaluation of neoplastic patients. However, evidence from large clinical randomized studies is lacking. Therefore, phase II and III trials are warranted in order to clarify the real impact of nutritional interventions on clinical outcomes.

## 4. Materials and Methods

All the abstracts presented both at the 2020 ASCO and ESMO World GI meetings were considered using the dedicated portals: https://meetinglibrary.asco.org and https://www.sciencedirect.com/journal/annals-of-oncology (Annals of Oncology Open Archive). Research terms were "nutrition" AND "gastrointestinal cancers" OR "gastrointestinal tumors" for ASCO abstracts, while "nutrition" was used for the ESMO World GI abstract collection. Only abstracts containing prospective or retrospective series concerning nutritional parameters, assessment/screening tools or interventions were considered. All abstracts concerning diseases different from GIC were excluded.

**Funding:** This research received no external funding

**Conflicts of Interest:** The authors declare no conflict of interest.

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
