# Peer review of "Nutritional Assessment in Gastrointestinal Tumors: News from the 2020 ASCO and ESMO World GI Meetings"

_gastrointestdisord, doi:10.3390/gidisord2030028_

Round 1

Reviewer 1 Report

The present work is an important addition to the existing knowledge in nutritional assessment in gastrointestinal tumors. Relevant news from the main oncologic meetings, ASCO and ESMO World GI, in 2020 have been summarized. In particular, new findings were assigned to specific tumor entities and adequately processed. The new findings from the two meetings will make a valuable contribution to the further treatment of patients at risk for malnutrition.

Great introduction. Clear arranged results. Well organised tables. Congratulations to this interestign work.

Cave: Page 4/96 2x with with

Author Response

Thanks to reviewer one for the kind comments on our paper. We truly appreciated them.

Reviewer 2 Report

this descriptive review aims at collecting the most recent advances in nutritional assessment of patients with Gastrointestinal cancer. In particular, 12 abstracts presented at the 2020 ASCO and ESMO World GI meetings are discussed.

The intention of this paper was to examine the most recent news concerning Nutritional assessment in gastrointestinal tumors.

As a matter of fact , the revised abstracts are highly inhomogeneous and deal with several assessment tools which mostly are unappropriated.

if you want to discuss nutritional assessment, you have to keep in mind that The tools to be employed for a screening  purpose include data obtained from the medical history, dietary intake, biochemical markers and, anthropometric data.

Talking about assessment, The presence of two or more of the following signs support a diagnosis of malnutrition: Insufficient energy intake (<75% of the individual's requirement), Involuntary weight loss (>5% in 3 months), Loss of subcutaneous fat mass, Loss of muscle mass, Generalized edema (which may conceal weight loss), Reduction in muscular function (eg, grip strength).

This review is a melting pot of extremely different assessments and protocols not always appropriate or validated. the scientific value of this work is very poor as examines a very partial segment of the nutritional issues in cancer patients mixing nutritional or metabolic markers, without giving a real message.

Author Response

We are sorry to understand that the aim of our review was not fully understood. This review wants to discuss the latest advances in the nutritional field of GI tumours presented at two international oncology meetings. We are fully aware that this is not a comprehensive review on the topic, and we stated it clearly in the introduction (lines 73-76).

Reviewer 3 Report

I would like to commend the authors for putting together an informative review which provides a summary of the most recent literature regarding nutritional assessment. Some minor points which might be worth addressing include: 

-Is it worth providing a definition of cachexia in the introduction eg as per Fearon et al. Lancet Oncol 2011

-Please expand what you mean by the term host-phagocitic syndrome in line 48

-In line 52 remove the word strong and leave it as 'considering the influence of optimal nutritional status on clinical outcomes'

-Line 59 where it says 'Garla', is this a reference that needs to be added?

-The authors suggest that the most appropriate are the SGA and PG-SA in line 62. Is this because they combine data from medical history and physical changes? If this is the rationale for them being the most appropriate please rephrase eg As the SGA and PG-SA combine medical history and physical changes, they are considered to be the most useful.

-Please correct in line 74: ASCO stands for American Society of Clinical Oncology

-Line 103 please change laboratories to laboratory parameters 

Author Response

Thanks for your valuable hints. We accepted your suggestions and tried to improve the text accordingly.

-Is it worth providing a definition of cachexia in the introduction eg as per Fearon et al. Lancet Oncol 2011. A definition of cachexia is already present in the introduction: “Cancer cachexia is a host-phagocitic syndrome with a progressive decline of skeletal muscle, with or without fat loss. It leads to worsening functional impairment, reduced efficacy of chemotherapy (CT), increased toxicities and a death rate up to 20% of all cancer-related deaths [4]” (lines 48-51)

-Please expand what you mean by the term host-phagocitic syndrome in line 48. In the text we modified this sentence in order to make it clearer: “Cancer cachexia is complex syndrome characterized by autophagy with a progressive decline of skeletal muscle” (lines 48-49).

-In line 52 remove the word strong and leave it as 'considering the influence of optimal nutritional status on clinical outcomes'. Thanks for the hint, we modified the text as you suggested.

-Line 59 where it says 'Garla', is this a reference that needs to be added? Sorry for the oversight. The reference is the number 1: Garla, P.; Waitzberg, D.L.; Tesser, A. Nutritional Therapy in Gastrointestinal Cancers. Gastroenterol Clin North Am 2018, 47, 231-242, doi:10.1016/j.gtc.2017.09.009.

-The authors suggest that the most appropriate are the SGA and PG-SA in line 62. Is this because they combine data from medical history and physical changes? If this is the rationale for them being the most appropriate please rephrase eg As the SGA and PG-SA combine medical history and physical changes, they are considered to be the most useful. The rephrased sentence is: “As the Subjective Global Assessment (SGA) and the Patient-Generated Subjective Global Assessment (PG-SGA) combine medical history (weight change, dietary intake change, gastrointestinal symptoms and functional capacity changes) together with physical changes (subcutaneous fat loss, muscle wasting, ascites). they are considered to be the most useful.” (lines 62-65).

-Please correct in line 74: ASCO stands for American Society of Clinical Oncology. Sorry for the oversight. We modified it.

Reviewer 4 Report

The paper is interesting and well-written. I just suggest to comment deeper in the Discussion the recent findings on the prognostic role of sarcopenia and nutritional assessment in general in main GI cancer.

For example, sarcopenia was found to correlate to better treatment outcomes in PC patients treated with EUS-guided celiac plexus neurolysis (PMID 32611849) or with standard chemotherapy (PMID 32883372).

Please explain also the major tools used to measure and define sarcopenia (along with their advantages and pitfalls).

Author Response

Thanks for your valuable hints. We accepted your suggestions and tried to improve the text accordingly.

- I just suggest to comment deeper in the Discussion the recent findings on the prognostic role of sarcopenia and nutritional assessment in general in main GI cancer.

For example, sarcopenia was found to correlate to better treatment outcomes in PC patients treated with EUS-guided celiac plexus neurolysis (PMID 32611849) or with standard chemotherapy (PMID 32883372).

We added the two suggested references (25 and 26). Therefore, we added “Sarcopenia has been described as a negative prognostic factor in patients with PC undergoing endoscopic ultrasound-guided celiac plexus neurolysis [25]. Moreover, a decrease in SMI within two months after the initiation of treatment with FOLFIRINOX for PC was associated with a significantly shorter OS [26].” Lines 241-245

Please explain also the major tools used to measure and define sarcopenia (along with their advantages and pitfalls).

We added the requested information. “Conventionally, cutoffs for skeletal muscle measurement at the level of the third lumbar vertebra (L3) are used (SMI). However, L3 evaluation is not always included in computed tomography scans. For these cases, alternative levels may be L2-L4-L5-L1 or thoracic vertebrae T12-T11-T10 [24].” Lines 238-241